# Palliative Gastrectomy Improves the Survival of Patients with Metastatic Early-Onset Gastric Cancer: A Retrospective Cohort Study

**Hang An, Peng-Yuan Wang and Yu-Cun Liu \***

Department of General Surgery, Peking University First Hospital, Beijing 100034, China
\* Correspondence: 2011110327@bjmu.edu.cn; Tel.: +86-18301451688

**Abstract: Background**: Recent studies have found that patients with incurable gastric cancer might benefit from palliative gastrectomy, but the impact of palliative gastrectomy on metastatic early-onset gastric cancer (mEOGC) patients remains unclear. **Methods**: We analyzed mEOGC patients enrolled in the Surveillance, Epidemiology, and End Results registry from January 2004 to December 2018. Propensity score matching (PSM) analysis with 1:1 matching and the nearest-neighbor matching method were used to ensure well-balanced characteristics between the groups of patients with palliative gastrectomy and those without surgery. Kaplan–Meier survival analysis and Cox proportional hazards regression models were used to evaluate the overall survival (OS) and cause-specific survival (CSS) risk with corresponding 95% confidence intervals (CIs). **Results**: Of 3641 mEOGC patients, 442 (12.1%) received palliative gastrectomy. After PSM, 596 patients were included in the analysis, with 298 in each group. For the matched cohort, the median survival was 8 months, and the 5-year survival was 4.0%. The median OS of mEOGC patients undergoing palliative gastrectomy was significantly longer than that of patients without surgery (13 months vs. 6 months, $p < 0.001$), and palliative gastrectomy remained an independent protective factor after adjusting for confounders (HR 0.459, 95% CI 0.382–0.552, $p < 0.001$), and the protective effect was robust in the subgroup analysis. Similar results were indicated in CSS. Stratified analyses by treatment modality also warranted the superiority of palliative-gastrectomy-based treatment in improving OS and CSS. **Conclusions**: mEOGC patients with palliative gastrectomy had a significantly longer survival time than patients without surgery. Exploratory analysis confirmed that surgery-based therapy modality was superior in improving survival time.

**Keywords:** metastasis; early-onset gastric cancer; palliative gastrectomy; survival; gastric cancer

## 1. Introduction

Gastric cancer is a global health problem, with approximately one million new cases per year [1]. Despite its worldwide decline in incidence and mortality over the past 5 decades, gastric cancer remains the fourth leading cause of cancer death, accounting for an estimated 769 thousand deaths each year [1]. Approximately one-third of patients are diagnosed with distant metastases at the first clinic visit [2]. For patients with distant metastases, systemic therapy is recommended as the first line of care by the National Comprehensive Cancer Network (NCCN) guideline [3]. However, despite chemotherapy and/or molecularly targeted biological therapy, the 5-year survival rate of patients with advanced gastric cancer rarely exceeds 5%, and the median survival time is usually considered to be less than one year [4,5]. More than half of patients with advanced gastric cancer will suffer from tumor-related adverse events, including gastrointestinal obstruction, intractable hemorrhage, and gastric perforation at the end stage of disease [4,6]. Emergency surgical intervention will be implemented in life-threatening conditions. For patients with mild or asymptomatic symptoms, palliative gastrectomy is also prudent because of the high postoperative morbidity and unknown survival benefits. However, with advancements

in surgical technology and perioperative management, the surgical mortality rates for noncurative gastrectomy have decreased from 20% to 5% [2,4]. Recently, several studies found that patients with incurable gastric cancer might benefit from palliative gastrectomy in overall survival (OS) compared with palliative chemotherapy and supportive care [6–10]. However, most studies are based on retrospective data, which increases the risk of substantial selection bias when selecting candidates for palliative gastrectomy.

Epidemiological investigations have demonstrated that gastric cancer is more common in elderly individuals, with the highest incidence of reported cases occurring at 55–80 years old [11]. Thanks to the effective treatment of Helicobacter pylori and the popularization of healthy lifestyles in the elderly population over the past few decades, the proportion of traditional late-onset gastric cancer (LOGC) has decreased, whereas the proportion of early-onset gastric cancer (EOGC) has increased relatively by year. EOGC (≤50 years old) accounts for 4.6–12.5% of the overall gastric cancer population according to the latest research [3,12–14]. Mounting studies have confirmed that EOGC patients have unique clinicopathological characteristics and genetic heterogeneity when compared with LOGC patients. In contrast to those with LOGC, patients with EOGC contained a larger proportion of signet ring cell carcinoma and a higher frequency of poor differentiation [12,13,15]. The pathogenesis of EOGC has less to do with the environment but is more related to gene mutation since CDH1 and RhoA gene mutations are more universal in younger patients [16,17]. Furthermore, young patients are more likely to develop peritoneal metastasis at primary diagnosis, and the prognosis of advanced EOGC is worse than that of elderly patients, even with standard treatment [15]. The significant clinicopathological and genetic differences between EOGC and LOGC suggest that the optimal therapeutic strategy for EOGC may vary from that for LOGC. Although recent studies have investigated the effect of palliative gastrectomy on the survival of advanced gastric cancer, the majority of these studies focused on patients aged over 50 years old. The survival advantage associated with primary tumor resection for metastatic EOGC (mEOGC) is still unclear. Hence, we conducted this retrospective study based on the Surveillance, Epidemiology, and End Results Program (SEER) to explore the effect of palliative gastrectomy on the survival of metastatic EOGC patients.

## 2. Materials and Methods

### 2.1. Database and Patients

This retrospective study was based on the Surveillance, Epidemiology, and End Results Program (SEER) (Incidence—SEER Research Plus Data, 18 Registries, November 2020 Sub (2000–2018)), released in April 2021. The SEER Program is an authoritative source for cancer statistics in the United States and currently collects data on cancer incidence and survival from population-based cancer registries, covering approximately 48% of the US population [18]. We obtained access to the SEER database using the ID number 19568-Nov2021 and used SEER*Stat software (version 8.4.0.1) to extract clinicopathologic and survival information. As patient data identified from the database were deidentified and available to the public for research purposes, the ethical approval of the present study was waived by the local ethics committee.

Due to database limitations, our analysis was restricted to samples from 1 January 2004 to 31 December 2018. At present, there is no specific definition for EOGC, and the cutoff age for EOGC in previous studies ranges from 30 to 50 years old [12,14,19]. Therefore, we defined EOGC patients as patients diagnosed with gastric cancer primarily before 50 years old in this study.

We enrolled patients according to the following criteria: (1) aged ≤ 50 years old; (2) diagnosed with primary gastric malignant tumor only; (3) confirmed synchronous distant metastasis at the first clinical visit; (4) confirmed epithelial carcinoma by pathological examination; (5) the status of palliative gastrectomy was known; and (6) the survival status was known. Patients were excluded if (1) surgery of the metastatic site had been performed

or was unknown; (2) surgery of the primary site was reported as local tumor destruction or damage; or (3) the survival status was 0 months.

The following parameters were collected from the database: age at diagnosis, sex, race, marital status, year of diagnosis, primary site of the tumor (cardia, non-cardia, overlapping lesion of stomach or unknown), tumor differentiation grade, histology (signet ring cell carcinoma, other adenocarcinoma or non-adenocarcinoma), tumor size (≤5 cm, >5 cm or unknown), T stage unified by the eighth edition of the American Joint Committee on Cancer (AJCC) staging manual [20], receipt of radiation, receipt of chemotherapy, overall survival (OS) in months (defined as the time interval from diagnosis to death from any cause), and cause-specific survival (CSS) in months (defined as the time interval from the diagnosis to death caused by gastric carcinoma). The receipt of surgery was defined as palliative primary tumor resection (palliative gastrectomy), including near-total or total gastrectomy and gastrectomy with a resection in continuity with the resection of other organs (RX Summ–Surg Prim Site (1998+), codes 30–33, 40–42, 50–52, 60–63, and 80). Specifically, only patients diagnosed after 2010 were assessed for synchronous metastatic patterns (including metastasis to the liver, lung, bone and brain), while the synchronous metastatic patterns of patients diagnosed before 2010 were classified as unknown. Since we enrolled patients with metastatic gastric cancer undergoing palliative tumor resection, an accurate pathological N-stage was unreliable, and the N-stage was not analyzed in this study arbitrarily.

## 2.2. Propensity Score Matching

Since this study was a nonrandomized retrospective study, one-to-one propensity score matching (PSM) was used to reduce the unwanted effect of treatment selection bias [21–23]. Propensity scores were established by a multivariable logistic regression model to calculate each patient's probability of receiving palliative surgery, in which the dependent variable was a binary indicator. Baseline variables with a *p*-value < 0.05 in the univariable logistic regression model were used to generate the propensity score. Considering that there was insufficient evidence of the pathological N stage, we did not match the N stage in our study. Patients who received palliative gastrectomy were matched to those who did not receive surgery by nearest-neighbor matching without replacement, with a minimum caliper of 0.10 [24]. The balance of baseline variables was assessed using the standardized mean difference (SMD), which was calculated as the mean difference (MD) divided by the standard deviation (SD) before and after PSM. Characteristics were considered imbalanced if either SMD > $1.96 \times \sqrt{(n1 + n2)/(n1 \times n2)}$ or *p*-values < 0.05 [21,25].

## 2.3. Statistical Analysis

For continuous variables with a normal distribution, descriptive statistics are expressed as the mean (standard deviation). For continuous variables with a non-normal distribution, descriptive statistics are expressed as the median (interquartile range). For classified variables, descriptive statistics are expressed as absolute numbers (proportions). Group comparisons of continuous variables were performed using Student's *t* test or the Mann–Whitney test, while categorical variables were compared with Pearson's chi-square test or Fisher's exact test. The primary endpoints of this study were OS and CSS. Correlations between various factors and the OS/CSS of mEOGC patients were assessed by univariate and multivariate Cox proportional hazards regression analyses. Variables with *p* < 0.05 in univariable analysis were adjusted as confounders in the multivariable analysis, and the results for significant prognostic factors were expressed as the hazard ratio (HR) for each category and its 95% confidence interval (CI).

Subgroup analyses of the OS and CSS between patients with or without palliative surgery were performed with stratification of age, sex, race, primary site, tumor differentiation grade, histology, tumor size, and T stage. Treatment by covariate interactions were assessed separately with Cox proportional hazards models for each subgroup factor. Furthermore, survival curves of OS and CSS of different treatment modalities (including no therapy, surgery only, radiation only, chemotherapy only, surgery + radiation,

surgery + chemotherapy, radiation + chemotherapy, and surgery + radiation + chemotherapy) were developed, and comparisons between different interventions were evaluated with multivariable Cox proportional hazards models after adjusting for the confounding variables mentioned above.

All analyses were performed with IBM SPSS Statistics 26.0 software (IBM, Armonk, NY, USA) and R version 4.1.3 (www.r-project.org, accessed on 1 June 2022), and two-tailed *p*-values < 0.05 were assessed as statistically significant.

## 3. Results

### 3.1. Baseline Characteristics

A total of 3641 patients with a mean age at diagnosis of 41.7 years old and a male predominance of 61.7% met our selection criteria (Table 1, Figure 1). The majority of patients with metastatic early-onset gastric cancer (mEOGC) were White (70.4%), followed by Asian or Pacific Islander (14.4%). The anatomic site of the tumor at diagnosis was mostly non-cardia (41.0%). The major pathological type of mEOGC was adenocarcinoma (95.8%), while signet ring cell carcinoma accounted for more than one-third. In addition, 64.4% of mEOGC patients were diagnosed as either poorly differentiated or undifferentiated tumors. Among these patients, 442 (12.1%) patients underwent palliative gastrectomy, whereas 3199 (87.9%) patients did not, and the proportions of patients receiving radiation and chemotherapy were 17.5% and 78.4%, respectively (Table 1). For CSS, 3042 (83.5%) patients died from gastric cancer, while 599 (16.5%) patients were alive or died of other causes at the last follow-up.

**Table 1.** Baseline data in the unmatched cohort.

| | Overall (*n* = 3641) | Non-Surgery (*n* = 3199) | Surgery (*n* = 442) | *p*-Value | Standardized Mean Difference |
|---|---|---|---|---|---|
| Age, year | 41.7 ± 7.1 | 41.6 ± 7.1 | 41.9 ± 6.7 | 0.483 | 0.036 |
| Sex | | | | 0.688 | 0.023 |
| Female | 1395 (38.3%) | 1230 (38.4%) | 165 (37.3%) | | |
| Male | 2246 (61.7%) | 1969 (61.6%) | 277 (62.7%) | | |
| Race | | | | **<0.001** | **0.225** |
| White | 2564 (70.4%) | 2283 (71.4%) | 281 (63.6%) | | |
| Black | 461 (12.7%) | 397 (12.4%) | 64 (14.5%) | | |
| American Indian/Alaska Native | 70 (1.9%) | 66 (2.1%) | 4 (0.9%) | | |
| Asian or Pacific Islander | 526 (14.4%) | 435 (13.6%) | 91 (20.6%) | | |
| Unknown | 20 (0.5%) | 18 (0.6%) | 2 (0.5%) | | |
| Marital status | | | | 0.021 | 0.161 |
| Married | 2013 (55.3%) | 1739 (54.4%) | 274 (62.0%) | | |
| Single | 1157 (31.8%) | 1042 (32.6%) | 115 (26.0%) | | |
| Divorced/Widowed/Separated | 321 (8.8%) | 285 (8.9%) | 36 (8.1%) | | |
| Unknown | 150 (4.1%) | 133 (4.2%) | 17 (3.8%) | | |
| Year of diagnosis | 2011 (2007, 2015) | 2011 (2007, 2015) | 2009 (2005, 2013) | **<0.001** | **0.401** |
| Primary site | | | | **<0.001** | **0.493** |
| Cardia | 955 (26.2%) | 901 (28.2%) | 54 (12.2%) | | |
| Non-cardia | 1491 (41.0%) | 1237 (38.7%) | 254 (57.5%) | | |
| Overlapping lesion of stomach | 419 (11.5%) | 355 (11.1%) | 64 (14.5%) | | |
| Unknown | 776 (21.3%) | 706 (22.1%) | 70 (15.8%) | | |
| Tumor size | | | | **<0.001** | **1.172** |
| ≤5 cm | 644 (17.7%) | 512 (16.0%) | 132 (29.9%) | | |
| >5 cm | 602 (16.5%) | 391 (12.2%) | 211 (47.7%) | | |
| Unknown | 2395 (65.8%) | 2296 (71.8%) | 99 (22.4%) | | |
| T stage | | | | **<0.001** | **1.507** |
| T1/ T2 | 660 (18.1%) | 637 (19.9%) | 23 (5.2%) | | |
| T3/ T4 | 1423 (39.1%) | 1024 (32.0%) | 399 (90.3%) | | |
| Tx | 1558 (42.8%) | 1538 (48.1%) | 20 (4.5%) | | |

**Table 1.** *Cont.*

| | Overall (*n* = 3641) | Non-Surgery (*n* = 3199) | Surgery (*n* = 442) | *p*-Value | Standardized Mean Difference |
|---|---|---|---|---|---|
| N stage | | | | **<0.001** | **1.472** |
| N0 | 1120 (30.8%) | 1068 (33.4%) | 52 (11.8%) | | |
| N1 | 1208 (33.2%) | 1062 (33.2%) | 146 (33.0%) | | |
| N2 | 221 (6.1%) | 94 (2.9%) | 127 (28.7%) | | |
| N3 | 168 (4.6%) | 64 (2.0%) | 104 (23.5%) | | |
| Nx | 924 (25.4%) | 911 (28.5%) | 13 (2.9%) | | |
| Histology | | | | **0.038** | 0.134 |
| Signet ring cell carcinoma | 1306 (35.9%) | 1127 (35.2%) | 179 (40.5%) | | |
| Other adenocarcinoma | 2181 (59.9%) | 1930 (60.3%) | 251 (56.8%) | | |
| Non-adenocarcinoma | 154 (4.2%) | 142 (4.4%) | 12 (2.7%) | | |
| Tumor differentiation grade | | | | | |
| I/II | 405 (11.1%) | 357 (11.2) | 48 (10.9) | **<0.001** | **0.490** |
| III/IV | 2343 (64.4%) | 1988 (62.1) | 355 (80.3) | | |
| Unknown | 893 (24.5%) | 854 (26.7) | 39 (8.8) | | |
| Metastasis to the liver [a] | | | | **<0.001** | **0.426** |
| Yes | 655 (18.0%) | 625 (19.5%) | 30 (6.8%) | | |
| No | 1483 (40.7%) | 1312 (41.0%) | 171 (38.7%) | | |
| Unknown | 1503 (41.3%) | 1262 (39.4%) | 241 (54.5%) | | |
| Metastasis to the lung [a] | | | | **<0.001** | **0.404** |
| Yes | 265 (7.3%) | 259 (8.1%) | 6 (1.4%) | | |
| No | 1859 (51.1%) | 1665 (52.0%) | 194 (43.9%) | | |
| Unknown | 1517 (41.7%) | 1275 (39.9%) | 242 (54.8%) | | |
| Metastasis to the bone [a] | | | | **<0.001** | **0.454** |
| Yes | 361 (9.9%) | 354 (11.1%) | 7 (1.6%) | | |
| No | 1774 (48.7%) | 1580 (49.4%) | 194 (43.9%) | | |
| Unknown | 1506 (41.4%) | 1265 (39.5%) | 241 (54.5%) | | |
| Metastasis to the brain [a] | | | | **<0.001** | **0.313** |
| Yes | 39 (1.1%) | 38 (1.2%) | 1 (0.2%) | | |
| No | 2088 (57.3%) | 1888 (59.0%) | 200 (45.2%) | | |
| Unknown | 1514 (41.6%) | 1273 (39.8%) | 241 (54.5%) | | |
| Radiation | | | | **<0.001** | **0.165** |
| Yes | 638 (17.5%) | 535 (16.7%) | 103 (23.3%) | | |
| No | 3003 (82.5%) | 2664 (83.3%) | 339 (76.7%) | | |
| Chemotherapy | | | | **0.014** | 0.124 |
| Yes | 2853 (78.4%) | 2527 (79.0%) | 326 (73.8%) | | |
| No | 788 (21.6%) | 672 (21.0%) | 116 (26.2%) | | |

Data are presented as mean ± standard deviation, n (%), or median (interquartile range). Standardized mean difference > 0.161 or two-tailed *p*-values < 0.05 were considered imbalanced between the two groups and expressed in bold. [a] The site of metastasis was only accessible for patients diagnosed after 2010, while the site of metastasis of patients diagnosed before 2010 was recorded as unknown.

The surgery group and non-surgery group had significantly different characteristics. Patients in the surgery group were more likely to have spouses (62.0% vs. 54.4%), and the proportion of Asian or Pacific Islander patients was higher (20.6% vs. 13.6%). In addition, the primary tumor in the surgery group was mostly located in the non-cardia (57.5% vs. 38.7%), and the tumor size was larger since the proportion of tumor size > 5 cm was 47.7% in the surgery group, while the proportion was 12.2% in the non-surgery group. Meanwhile, pathological examination indicated that signet ring cell carcinoma was more common in the surgery group (40.5% vs. 35.2%) (Table 1, Figure 2A). Patients in the surgery group were more likely to receive radiotherapy (23.3% vs. 16.7%) and were less likely to receive chemotherapy (73.8% vs. 79.0%) (Table 1).

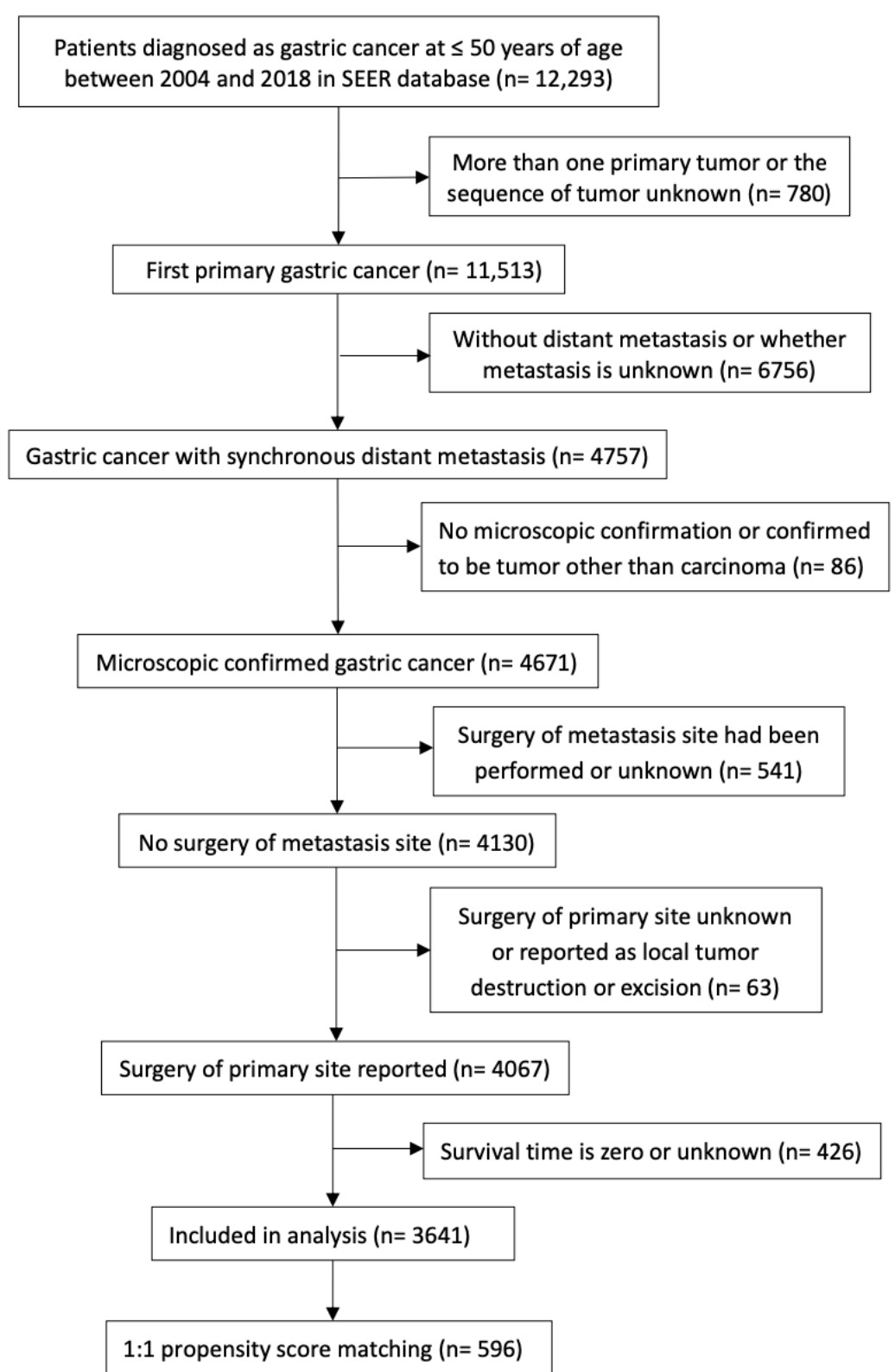

**Figure 1.** Flowchart. SEER = Surveillance, Epidemiology, and End Results Program.

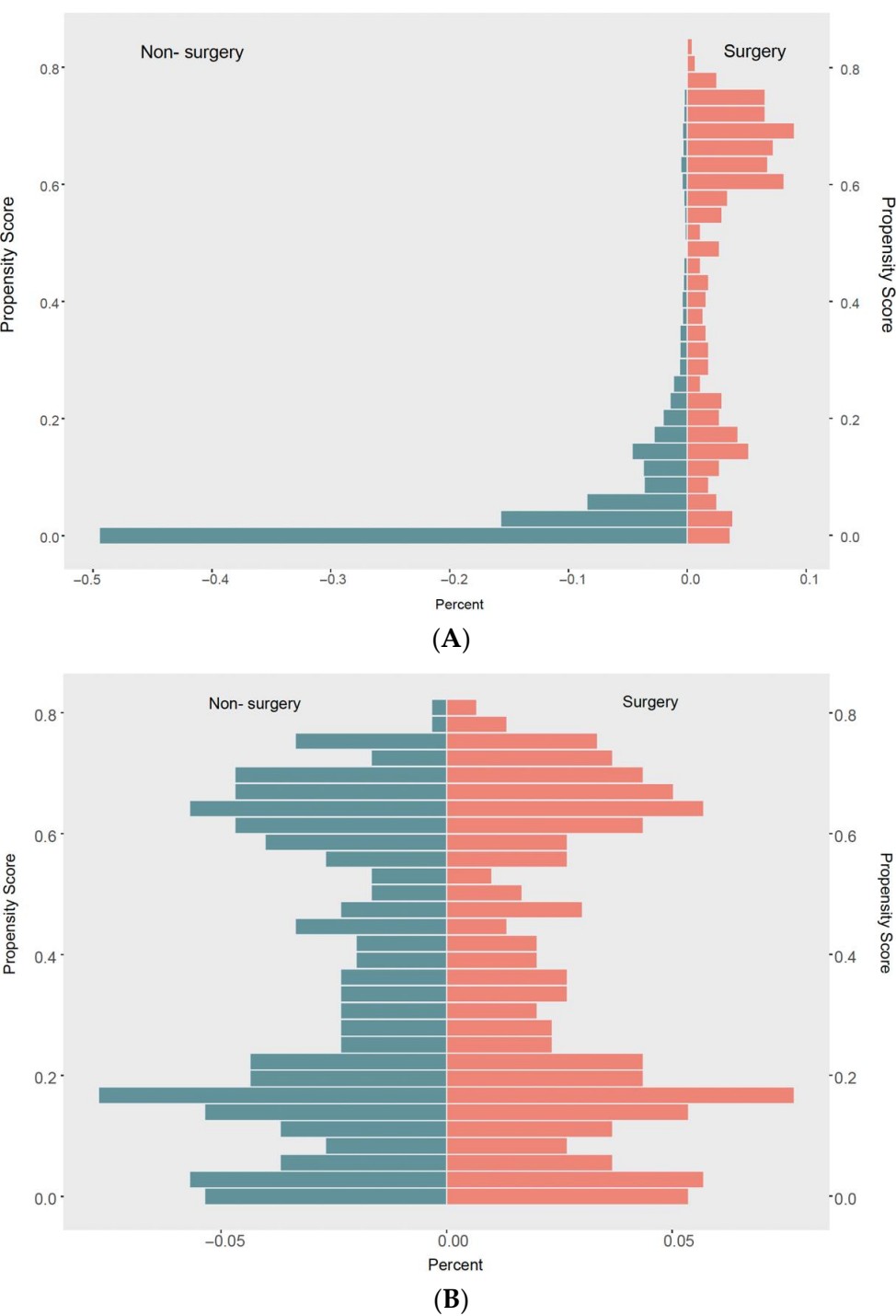

**Figure 2.** (**A**) Mirror histograms before matching. (**B**) Mirror histograms after matching.

To balance the baseline characteristics between the surgery and non-surgery groups, PSM analysis was conducted, and the variables used for matching were as follows: age, sex, race, marital status, year of diagnosis, primary site of the tumor, tumor size, T stage, histology, tumor differentiation grade, metastatic sites, receipt of radiation, and receipt of chemotherapy. After the PSM analysis, 596 mEOGC patients were 1:1 matched in surgery and non-surgery cohorts. Baseline characteristics between the two groups were well balanced after PSM, and the SMD between the two groups was below 0.161 ($1.96 \times \sqrt{(n1 + n2)/(n1 \times n2)}$) (Table 2, Figures 2B and 3).

**Table 2.** Baseline data in the propensity score matched cohort.

| | Overall (*n* = 596) | Non-Surgery (*n* = 298) | Surgery (*n* = 298) | *p*-Value | Standardized Mean Difference |
|---|---|---|---|---|---|
| Age, year | 41.8 ± 6.7 | 41.7 ± 6.6 | 41.8 ± 6.8 | 0.765 | 0.025 |
| Sex | | | | 0.357 | 0.082 |
| Female | 236 (39.6%) | 124 (41.6%) | 112 (37.6%) | | |
| Male | 360 (60.4%) | 174 (58.4%) | 186 (62.4%) | | |
| Race | | | | 0.982 | 0.052 |
| White | 392 (65.8%) | 194 (65.1%) | 198 (66.4%) | | |
| Black | 79 (13.3%) | 40 (13.4%) | 39 (13.1%) | | |
| American Indian/Alaska Native | 7 (1.2%) | 4 (1.3%) | 3 (1.0%) | | |
| Asian or Pacific Islander | 113 (19.0%) | 57 (19.1%) | 56 (18.8%) | | |
| Unknown | 5 (0.8%) | 3 (1.0%) | 2 (0.7%) | | |
| Marital status | | | | 0.994 | 0.024 |
| Married | 357 (59.9%) | 178 (59.7%) | 179 (60.1%) | | |
| Single | 165 (27.7%) | 83 (27.9%) | 82 (27.5%) | | |
| Divorced/Widowed/Separated | 57 (9.6%) | 29 (9.7%) | 28 (9.4%) | | |
| Unknown | 17 (2.9%) | 8 (2.7%) | 9 (3.0%) | | |
| Year of diagnosis | 2009 (2006, 2014) | 2010 (2006, 2014) | 2009 (2006, 2013) | 0.548 | 0.053 |
| Primary site | | | | 0.779 | 0.086 |
| Cardia | 100 (16.8%) | 46 (15.4%) | 54 (18.1%) | | |
| Non-cardia | 315 (52.9%) | 158 (53.0%) | 157 (52.7%) | | |
| Overlapping lesion of stomach | 82 (13.8%) | 44 (14.8%) | 38 (12.8%) | | |
| Unknown | 99 (16.6%) | 50 (16.8%) | 49 (16.4%) | | |
| Tumor size | | | | 0.981 | 0.016 |
| ≤5 cm | 174 (29.2%) | 87 (29.2%) | 87 (29.2%) | | |
| >5 cm | 226 (37.9%) | 114 (38.3%) | 112 (37.6%) | | |
| Unknown | 196 (32.9%) | 97 (32.6%) | 99 (33.2%) | | |
| T stage | | | | 0.680 | 0.072 |
| T1/ T2 | 43 (7.2%) | 20 (6.7%) | 23 (7.7%) | | |
| T3/ T4 | 508 (85.2%) | 253 (84.9%) | 255 (85.6%) | | |
| Tx | 45 (7.6%) | 25 (8.4%) | 20 (6.7%) | | |
| N stage | | | | <0.001 | 1.119 |
| N0 | 139 (23.3%) | 98 (32.9%) | 41 (13.8%) | | |
| N1 | 220 (36.9%) | 116 (38.9%) | 104 (34.9%) | | |
| N2 | 94 (15.8%) | 15 (5.0%) | 79 (26.5%) | | |
| N3 | 70 (11.7%) | 8 (2.7%) | 62 (20.8%) | | |
| Nx | 73 (12.2%) | 61 (20.5%) | 12 (4.0%) | | |
| Histology | | | | 0.410 | 0.110 |
| Signet ring cell carcinoma | 242 (40.6%) | 120 (40.3%) | 122 (40.9%) | | |
| Other adenocarcinoma | 325 (54.5%) | 160 (53.7%) | 165 (55.4%) | | |
| Non-adenocarcinoma | 29 (4.9%) | 18 (6.0%) | 11 (3.7%) | | |
| Tumor differentiation grade | | | | 0.659 | 0.075 |
| I/II | 72 (12.1%) | 33 (11.1%) | 39 (13.1%) | | |
| III/IV | 449 (75.3%) | 225 (75.5%) | 224 (75.2%) | | |
| Unknown | 75 (12.6%) | 40 (13.4%) | 35 (11.7%) | | |
| Metastasis to the liver [a] | | | | 0.989 | 0.012 |
| Yes | 55 (9.2%) | 28 (9.4%) | 27 (9.1%) | | |
| No | 230 (38.6%) | 115 (38.6%) | 115 (38.6%) | | |
| Unknown | 311 (52.2%) | 155 (52.0%) | 156 (52.3%) | | |
| Metastasis to the lung [a] | | | | 0.986 | 0.014 |
| Yes | 12 (2.0%) | 6 (2.0%) | 6 (2.0%) | | |
| No | 272 (45.6%) | 137 (46.0%) | 135 (45.3%) | | |
| Unknown | 312 (52.3%) | 155 (52.0%) | 157 (52.7%) | | |

**Table 2.** *Cont.*

| | Overall (*n* = 596) | Non-Surgery (*n* = 298) | Surgery (*n* = 298) | *p*-Value | Standardized Mean Difference |
|---|---|---|---|---|---|
| Metastasis to the bone [a] | | | | 0.986 | 0.014 |
| Yes | 14 (2.3%) | 7 (2.3%) | 7 (2.3%) | | |
| No | 272 (45.6%) | 137 (46.0%) | 135 (45.3%) | | |
| Unknown | 310 (52.0%) | 154 (51.7%) | 156 (52.3%) | | |
| Metastasis to the brain [a] | | | | 0.845 | 0.048 |
| Yes | 3 (0.5%) | 2 (0.7%) | 1 (0.3%) | | |
| No | 282 (47.3%) | 141 (47.3%) | 141 (47.3%) | | |
| Unknown | 311 (52.2%) | 155 (52.0%) | 156 (52.3%) | | |
| Radiotherapy | | | | >0.999 | 0.008 |
| Yes | 139 (23.3%) | 69 (23.2%) | 70 (23.5%) | | |
| No | 457 (76.7%) | 229 (76.8%) | 228 (76.5%) | | |
| Chemotherapy | | | | 0.238 | 0.105 |
| Yes | 463 (77.7%) | 238 (79.9%) | 225 (75.5%) | | |
| No | 133 (22.3%) | 60 (20.1%) | 73 (24.5%) | | |

Data are presented as mean ± standard deviation, n (%), or median (interquartile range). [a] The site of metastasis was only accessible for patients diagnosed after 2010, while the site of metastasis of patients diagnosed before 2010 was recorded as unknown.

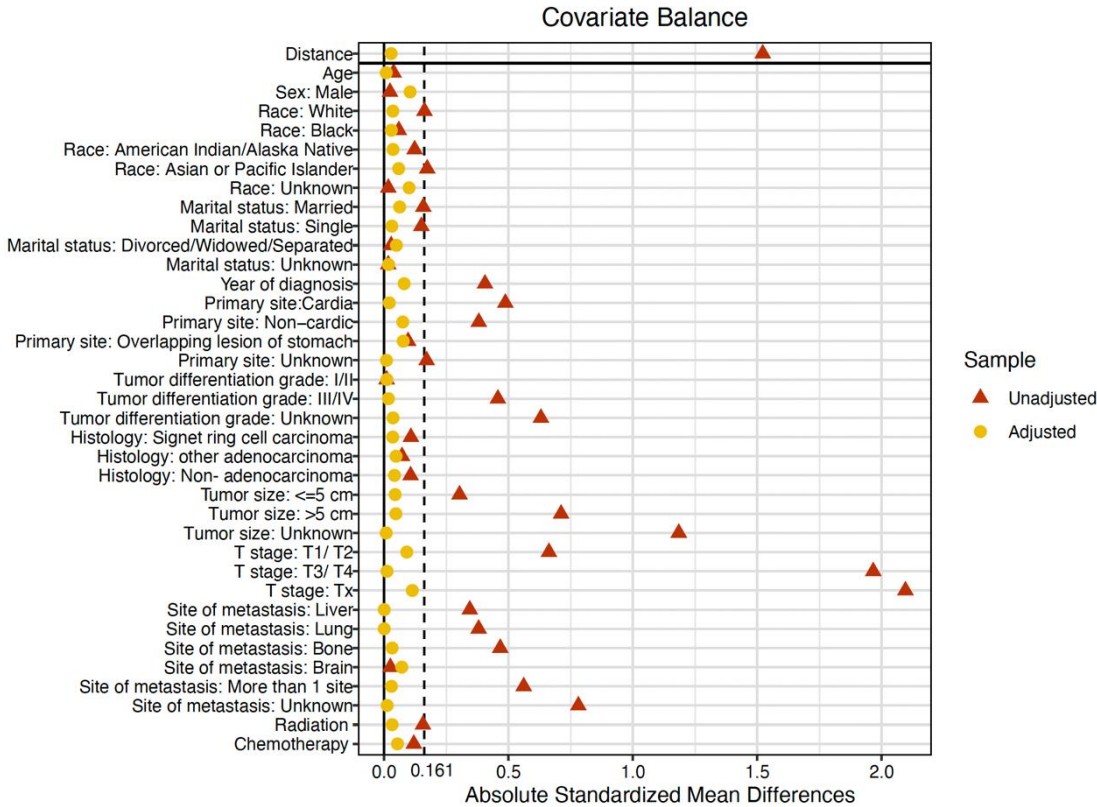

**Figure 3.** Standardized mean differences before and after matching.

### 3.2. Survival Analysis of Palliative Gastrectomy

For the matched cohort, the median survival was 8 months, and the 5-year survival was 4.0%. In the matched cohort, patients undergoing palliative gastrectomy had a significantly longer median OS time than patients without surgery (13 months [95% CI 11–15 months] vs. 6 months [95% CI 5–7 months], *p* < 0.001) (Figure 4A). A similar result was demonstrated in CSS (13 months [95% CI 11–16 months] vs. 6 months [95% CI 5–7 months], *p* < 0.001) (Figure 4B).

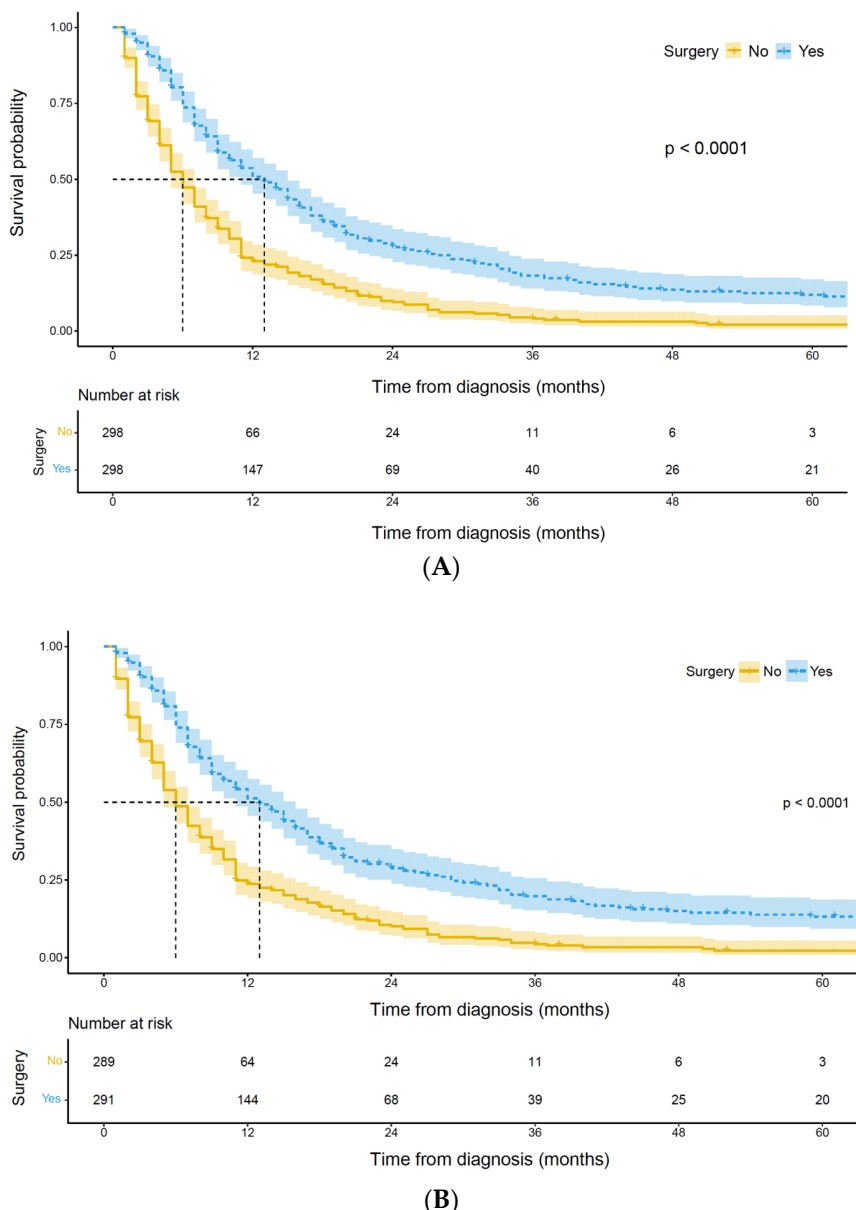

**Figure 4.** (**A**) Overall survival of patients with and without surgery. (**B**) Cause-specific survival of patients with and without surgery.

A univariate Cox proportional hazards model was applied to the matched population and indicated that characteristics including year of diagnosis, primary site of tumor, histology, T stage, metastasis to the liver, metastasis to the lung, metastasis to the bone, metastasis to the brain, receipt of chemotherapy, and receipt of surgery might be related to OS and CSS. The variables mentioned above were all included in the following multivariate Cox analysis. After adjusting for these confounding factors, the receipt of palliative gastrectomy remained an independent protective factor for OS (HR 0.459, 95% CI 0.382–0.552, $p < 0.001$) (Table 3). In addition, receipt of chemotherapy also indicated better OS (HR 0.640, 95% CI 0.512–0.801, $p < 0.001$). A similar result was demonstrated in CSS (Supplemental Table S1).

**Table 3.** Prognostic factors for overall survival.

| Variable | Univariable Cox Regression Analysis | | | Multivariable Cox Regression Analysis [a] | | |
|---|---|---|---|---|---|---|
| | Hazard Ratio | 95% CI | *p*-Value | Hazard Ratio | 95% CI | *p*-Value |
| Age, year | 1.000 | 0.987–1.013 | 0.981 | | | |
| Male sex | 0.915 | 0.767–1.092 | 0.326 | | | |
| Race | | | | | | |
| White | reference | | | | | |
| Black | 0.956 | 0.745–1.229 | 0.727 | | | |
| American Indian/Alaska Native | 1.445 | 0.683–3.056 | 0.335 | | | |
| Asian or Pacific Islander | 0.852 | 0.675–1.074 | 0.174 | | | |
| Unknown | 0.560 | 0.179–1.752 | 0.319 | | | |
| Marital status | | | | | | |
| Married | reference | | | | | |
| Single | 0.840 | 0.686–1.027 | 0.090 | | | |
| Divorced/Widowed/Separated | 0.839 | 0.621–1.135 | 0.254 | | | |
| Unknown | 1.018 | 0.615–1.683 | 0.946 | | | |
| Year of diagnosis | 0.972 | 0.951–0.993 | **0.008** | 0.980 | 0.940–1.023 | 0.355 |
| Primary site | | | | | | |
| Cardia | reference | | | reference | | |
| Non-cardia | 1.424 | 1.112–1.823 | **0.005** | 1.349 | 1.031–1.766 | **0.029** |
| Overlapping lesion of stomach | 1.517 | 1.099–2.093 | **0.011** | 1.381 | 0.982–1.942 | 0.064 |
| Unknown | 1.599 | 1.182–2.164 | **0.002** | 1.473 | 1.066–2.037 | **0.019** |
| Tumor differentiation grade | | | | | | |
| I/II | reference | | | reference | | |
| III/IV | 1.291 | 0.986–1.690 | 0.063 | | | |
| Unknown | 0.996 | 0.685–1.448 | 0.983 | | | |
| Histology | | | | | | |
| Signet ring cell carcinoma | reference | | | reference | | |
| Other adenocarcinoma | 0.810 | 0.678–0.969 | **0.021** | 0.854 | 0.706–1.031 | 0.101 |
| Non-adenocarcinoma | 0.586 | 0.370–0.927 | **0.022** | 0.579 | 0.355–0.942 | **0.028** |
| Tumor size | | | | | | |
| ≤5 cm | reference | | | | | |
| >5 cm | 1.063 | 0.862–1.312 | 0.568 | | | |
| Unknown | 1.115 | 0.893–1.391 | 0.336 | | | |
| T stage | | | | | | |
| T1/ T2 | reference | | | reference | | |
| T3/ T4 | 1.421 | 1.005–2.008 | **0.047** | 1.346 | 0.947–1.915 | 0.098 |
| Tx | 1.446 | 0.909–2.300 | 0.120 | 1.547 | 0.965–2.480 | 0.070 |
| Metastasis to the liver [b] | | | | | | |
| No | reference | | | reference | | |
| Yes | 1.146 | 0.822–1.598 | 0.420 | 1.133 | 0.791–1.623 | 0.496 |
| Unknown | 1.298 | 1.079–1.562 | **0.006** | 1.056 | 0.108–10.332 | 0.963 |
| Metastasis to the lung [b] | | | | | | |
| No | reference | | | reference | | |
| Yes | 1.933 | 1.024–3.647 | **0.042** | 3.228 | 1.634–6.378 | **0.001** |
| Unknown | 1.302 | 1.091–1.553 | **0.003** | 2.133 | 0.301–15.103 | 0.448 |

**Table 3.** *Cont.*

| Variable | Univariable Cox Regression Analysis | | | Multivariable Cox Regression Analysis [a] | | |
|---|---|---|---|---|---|---|
| | Hazard Ratio | 95% CI | *p*-Value | Hazard Ratio | 95% CI | *p*-Value |
| Metastasis to the bone [b] | | | | | | |
| No | reference | | | reference | | |
| Yes | 1.769 | 1.030–3.038 | **0.039** | 1.918 | 1.099–3.348 | **0.022** |
| Unknown | 1.299 | 1.088–1.551 | **0.004** | 0.703 | 0.068–7.316 | 0.768 |
| Metastasis to the brain [b] | | | | | | |
| No | reference | | | reference | | |
| Yes | 4.628 | 1.473–14.542 | **0.009** | 4.517 | 1.404–14.538 | **0.011** |
| Unknown | 1.277 | 1.072–1.521 | **0.006** | 0.675 | 0.093–4.903 | 0.698 |
| Receipt of radiation | | | | | | |
| No | reference | | | | | |
| Yes | 0.911 | 0.745–1.114 | 0.366 | | | |
| Receipt of chemotherapy | | | | | | |
| No | reference | | | reference | | |
| Yes | 0.727 | 0.591–0.894 | **0.003** | 0.640 | 0.512–0.801 | **<0.001** |
| Receipt of surgery | | | | | | |
| No | reference | | | reference | | |
| Yes | 0.494 | 0.414–0.590 | **<0.001** | 0.459 | 0.382–0.552 | **<0.001** |

CI = confidence interval. *p*-values in bold indicate < 0.05 and are considered as statistically significant. [a] Confounding factors (including year of diagnosis, primary site, histology, T stage, metastasis to the liver, metastasis to the lung, metastasis to the bone, metastasis to the brain, receipt of chemotherapy, receipt of surgery) were adjusted in multivariable Cox proportional hazards regression analysis. [b] The site of metastasis was only accessible for patients diagnosed after 2010, while the site of metastasis of patients diagnosed before 2010 was recorded as unknown.

Consistent with multivariable Cox proportional hazards regression analysis, subgroup analysis also confirmed that patients benefited from palliative gastrectomy in improving OS and CSS (Supplemental Figures S1 and S2). Specifically, patients suffering from poorly differentiated or undifferentiated tumors benefitted more from receiving palliative gastrectomy than patients with well-differentiated tumors (Supplemental Figures S1 and S2).

*3.3. Impact of Treatment Modality*

In the matched cohort, we stratified mEOGC patients into eight groups (no therapy, only surgery, only radiation, only chemotherapy, surgery + radiation, surgery + chemotherapy, radiation + chemotherapy, and surgery + radiation + chemotherapy) according to their treatment modality. The Kaplan–Meier survival analysis indicated that patients with surgery-based trimodality therapy (15 months, 95% CI 12–18 months) and patients with surgery combined with chemotherapy (14 months, 95% CI 10–17 months) had the longest median OS time, while patients who did not receive any therapy had the shortest median OS time (3 months, 95% CI 2–5 months) (Figure 5A). In addition, patients who only received surgery had a longer median OS time (9 months, 95% CI 6–19 months) than patients who only received chemotherapy (7 months, 95% CI 5–8 months) and patients who only received radiation (2 months, 95% CI 1–3 months) (Figure 5A). After adjusting for confounders (including year of diagnosis, primary site of tumor, histology, T stage, metastases to the liver, metastases to the lung, metastases to the bone, metastases to the brain), surgery-based treatment remained superior in improving OS (Supplemental Table S2). Similar results were also demonstrated in CSS (Figure 5B, Supplemental Table S2).

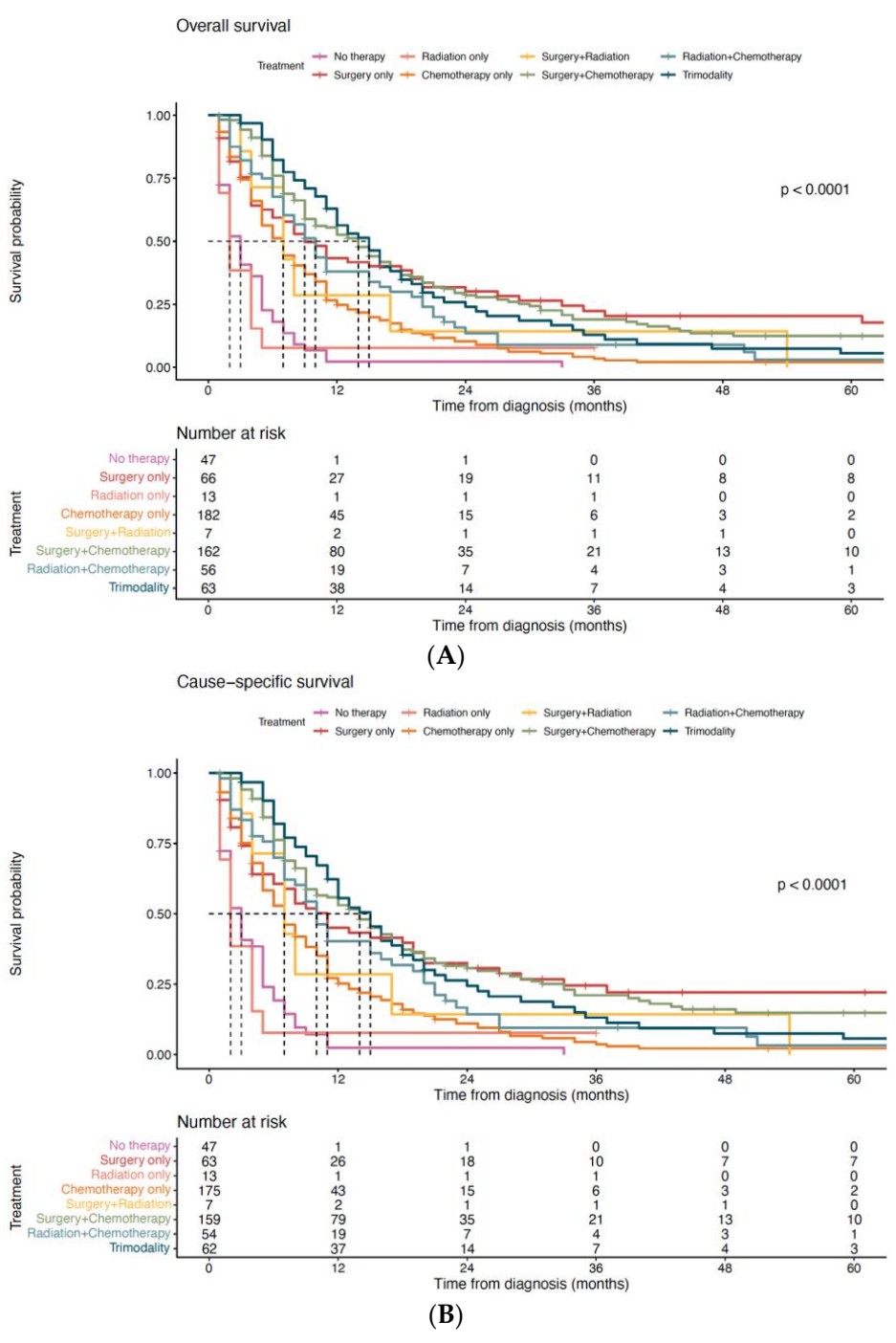

**Figure 5.** (**A**) The impact of treatment modalities on overall survival. (**B**) The impact of treatment modalities on cause-specific survival.

## 4. Discussion

In this population-based cohort study, we noticed that mEOGC patients had a remarkable survival improvement when they received palliative gastrectomy. In addition, exploratory analysis confirmed that patients who received surgery-based trimodality had the longest median OS/CSS time, indicating the superiority of palliative gastrectomy.

The utility of palliative surgery in patients with advanced gastric cancer remains the subject of intense debate. Histologically, palliative surgery was declined by surgeons due to high perioperative mortality [4]. However, along with the development of surgical techniques and perioperative care, the incidence of perioperative complications decreased, as well as that of perioperative mortality. A retrospective study of 151 patients found that

patients who underwent palliative surgery to alleviate tumor-related symptoms resumed oral feeding sooner and experienced markedly fewer vomiting and gastrointestinal bleeding events than patients undergoing non-resectional procedures [26]. An increasing number of studies have demonstrated that advanced gastric cancer patients receiving palliative surgery tend to have a survival advantage over patients without any treatment or receiving palliative chemotherapy alone [9,10,27–30]. However, this superiority of palliative surgery was not corroborated by a randomized controlled trial (RCT). REGATTA was the only phase III trial that enrolled 175 advanced gastric cancer patients with a single non-curable factor. Patients included in this study were given sufficient oral intake and were free of active bleeding from the gastric tumor, while patients suffering from tumor-related symptoms such as ingestive tract obstruction, stomach perforation, and gastric hemorrhage were excluded. This trial found that there was no difference between the overall survival of patients receiving gastrectomy plus chemotherapy and patients receiving chemotherapy alone (HR 1.09, 95% CI 0.78–1.52; one-sided $p$-value = 0.70) [31]. However, this trial was criticized for its limited power because it was terminated in advance due to the futility of gastrectomy plus chemotherapy in interim analysis and poor participant accrual. In addition, this research was also criticized for enrolling numerous participants with total gastrectomy who required the administration of oral chemotherapy treatment regimens at the same time. The latter limitation might impede the periodic treatment of chemotherapy for patients receiving gastrectomy. Furthermore, this trial did not demonstrate if patients with severe tumor-related complications benefit from palliative gastrectomy.

Although several studies have been carried out to determine the effect of palliative surgery on advanced gastric cancer patients, most have focused on elderly patients. Little is known about the role of palliative surgery in mEOGC. Previous studies suggested that there was a significant clinicopathological difference between EOGC and LOGC, and the incidence of distant metastasis at diagnosis was higher for EOGC patients [32–34]. Furthermore, the prognosis of young individuals with advanced or unresectable gastric cancer is worse than that of elderly patients (12 months vs. 17.5 months, $p < 0.001$), which might be attributed to the prevalence of signet ring cell carcinoma and poor differentiation [15]. A retrospective study of 46 mEOGC patients (aged $\leq$ 45 years old) found that palliative surgery (HR 0.212, 95% CI 0.088–0.513, $p = 0.001$) was a significant prognostic predictor after adjusting for confounders [15]. However, this conclusion was not sound enough due to insufficient sample size and potential selection bias. Our study was based on the SEER database, which covers approximately 48% of the US population and involves various races, further strengthening the viability of our conclusion [18]. We used PSM to balance the baseline characteristics and mirrored the real-world outcomes. After adjusting for confounders, palliative surgery remained an independent protective factor to improve OS (HR 0.479, 95% CI, 0.397–0.576, $p < 0.001$) and CSS (HR 0.486, 95% CI, 0.402–0.588, $p < 0.001$).

The underlying mechanism of how palliative gastrectomy improves the survival outcome of mEOGC patients is still unclear. There may be several reasons for the significant survival benefits of palliative gastrectomy. First, as mentioned above, palliative gastrectomy provides symptomatic relief and reduces tumor-related complications such as gastrointestinal bleeding, perforation, or obstruction in the end stage of the tumor [6]. In addition, compared with elderly patients, young patients usually have better physical performance and fewer preoperative comorbidities, so they are more tolerant of surgery and have a lower risk of perioperative complications and mortality [35,36]. Second, removing the primary site tumor can relieve the hypercatabolic state and reduce the overall tumor burden without impairing natural killer cell cytotoxicity for a long time, thereby facilitating durable systemic chemotherapy [4,37]. Some studies equally found that the addition of chemotherapy to surgery was associated with better survival, which may be attributed to a better response to chemotherapy after reduction in systemic tumor burden. Third, some previous studies discovered that primary tumor resection was associated with recovery of the immune system, leading to survival improvement. It is supposed that

surgery could probably reverse systemic inflammation and restore immune function [38]. In the era of individualized treatment, palliative gastrectomy should be regarded as a part of a comprehensive multidisciplinary treatment for patients with mEOGC.

There are several limitations in our study. First, as a retrospective study, although efforts have been made to reduce selection bias by utilizing the PSM method, there may be unobserved confounders. For example, we failed to collect information about the reason for the patients' surgery and whether this was emergency surgery. Since the prognosis of emergency surgery is worse than that of elective surgery in most cases, patients receiving emergency surgery might die within one month. However, these patients were excluded from this study, which might have exaggerated the impact of palliative gastrectomy on survival. Furthermore, the reasons that patients received palliative gastrectomy, or not, can also affect the patients' prognosis to some extent. Second, although we balanced the baseline characteristics by performing PSM, the sample size was reduced significantly, and not every patient could be matched. Hence, the results may only be applicable to a subset of matched patients. Third, and importantly, the pattern of metastases was only restricted to the liver, lung, bone, and brain, while the status of peritoneal metastasis (PM) was not accessible. PM was also a potential factor influencing the clinical decision of surgery and postoperative recovery. Previous studies have shown that PM is an independent risk factor for prognosis, and patients with PM are recommended to receive multimodal treatment, including systemic chemotherapy, surgery, and intraperitoneal chemotherapy [39]. Fourth, the impact on quality of life (QOL) remained unclear since the SEER database did not collect corresponding information. QOL is indispensable when weighing the advantages and disadvantages of therapy, especially for metastatic gastric cancer patients. A previous study used hospitalization-free survival (HFS) as an alternative parameter to evaluate QOL, and the results showed that palliative gastrectomy might not be harmful [40].

Therefore, further studies with larger samples are needed to verify the benefit of palliative gastrectomy in terms of survival and QOL.

## 5. Conclusions

The present study showed that mEOGC patients receiving palliative gastrectomy had a significantly longer survival than patients without surgery. The beneficial effect remained after adjusting for confounders and was robust in the subgroup analysis. Exploratory analysis confirmed that surgery-based therapy modality was superior in improving OS and CSS. However, this retrospective cohort study did not collect the status of peritoneal metastasis, which was also a potential factor influencing the clinical decision of surgery and postoperative recovery, making the superiority of palliative gastrectomy questionable. Furthermore, the impact on the quality of life after surgery was also unknown. Thus, further well-designed studies with larger samples are needed to verify the benefit of palliative gastrectomy in terms of survival and QOL.

**Supplementary Materials:** The following are available online at https://www.mdpi.com/article/10.3390/curroncol30090572/s1, Table S1: Prognostic factors for cause-specific survival; Table S2: The effect of treatment modalities on overall survival and cause-specific survival; Figure S1: Forest plot predefined subgroups based on overall survival; Figure S2: Forest plot predefined subgroups based on cause-specific survival.

**Author Contributions:** H.A.: Conceptualization, data curation, formal analysis, investigation, methodology, project administration, software, and writing—original draft; P.-Y.W.: data curation, methodology, visualization, and writing—review and editing; Y.-C.L.: conceptualization, data curation, formal analysis, methodology, supervision, validation, and writing—review and editing. All authors have read and agreed to the published version of the manuscript.

**Funding:** This research received no external funding.

**Institutional Review Board Statement:** This retrospective study was based on the Surveillance, Epidemiology, and End Results Program (SEER) database. As this program collects data from

population-based cancer registries with anonymous information and is available to the public for research purposes, the ethical approval of the present study was waived by the local ethics committee.

**Informed Consent Statement:** Not applicable.

**Data Availability Statement:** The raw data of this study are derived from the SEER database, which is a publicly available database. All detailed data included in the study are available upon request by contacting the corresponding author.

**Acknowledgments:** The authors gratefully acknowledge the SEER program for open access to their database.

**Conflicts of Interest:** The authors declare that they have no competing interest.

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
