# Peer review of "Palliative Gastrectomy Improves the Survival of Patients with Metastatic Early-Onset Gastric Cancer: A Retrospective Cohort Study"

_curroncol, doi:10.3390/curroncol30090572_

Round 1

Reviewer 1 Report

This manuscript is an original article that retrospectively investigated the effect of palliative gastrectomy on the survival of metastatic early-onset gastric cancer (EOGC) patients using propensity score matching analysis.

The authors showed the median OS of mEOGCs undergoing palliative gastrectomy was significantly longer than that of patients without surgery and that palliative gastrectomy remained an independent protective factor after adjusting for confounders.

This study was conducted well, and the methods are appropriate. The data are presented clearly. In general, this is a well-written paper that presents interesting data. The results will be of interest to clinicians in the field.

However, the following minor issues require clarification:

Minor

1.     (P7L189-193) These sentences should be placed right under Table 1.

2.     The status of peritoneal metastasis (PM) is also a potential factor influencing the clinical decision of surgery and postoperative recovery and PM is an independent risk factor for prognosis, as the authors commented in the limitation. I’m concerned that the lack of data regarding PM may be the greatest weakness in this study. I recommend that the authors modify the conclusion based on this concern.

Reviewer 2 Report

The authors investigated the survival benefit of palliative gastrectomy for metastatic early-onset gastric cancer (mEOGC) using propensity score matching (PSM) and concluded that surgery-based treatments were superior in improving survival time.

(Major problems)

1.       The authors described the distant metastatic sites about liver, lung, bone, and brain metastasis. These are all hematogenous metastases. As for lymph node metastasis, the authors did not investigate lymphatic metastasis because it was difficult to estimate by palliative gastrectomy. Even if it was not appropriate to estimate pathological findings, how about estimating by clinical findings? Additionally, the authors should investigate about peritoneal metastasis. Although the authors discuss about this point as a limitation, this seems to be a big defect of the data. Distant metastatic sites are described in Table1 and 2, however, that seems not to be enough. The authors should investigate and describe more about the distant metastasis, and if possible, match these factors during PSM.

2.       Some important prognostic factors such as performance status are not investigated in this study. Patients with good performance status tend to undergo surgery, and this may be associated with better survival. The authors used PSM to exclude bias, but they should match other factors.

3.       The distant metastatic sites of patients who were diagnosed before 2010 were classified as unknown. Why did not the authors exclude these patients?  

(Minor problems)

1.       In REGATTA study, only asymptomatic (i.e. eligible to oral intake and no active bleeding) patients were included.  This study does not deny the benefit of palliative gastrectomy for symptomatic patients, so the authors should discuss about this point.

2.       Were patients with synchronous cancer of other organs excluded in this study?

3.       In Table 3, both of hazard ratio about with and without lung metastasis were shown. I guess “no metastasis to the lung” should be a reference.

Round 2

Reviewer 1 Report

Thank you for revising your manuscript according to my suggestion. The revised manuscript is improved enough to be accepted. 

Reviewer 2 Report

The authors have added data and descriptions appropriately. I think the manuscript is acceptable in present form.